# Imaging Techniques to Study Plant Virus Replication and Vertical Transmission

**DOI:** 10.3390/v13030358

**Published:** 2021-02-25

**Authors:** María Amelia Sánchez Pina, Cristina Gómez-Aix, Eduardo Méndez-López, Blanca Gosalvez Bernal, Miguel A. Aranda

**Affiliations:** 1Centro de Edafología y Biología Aplicada del Segura (CEBAS-CSIC), Departamento de Biología del Estrés y Patología Vegetal, Grupo de Patología Vegetal, 30100 Murcia, Spain; fmendez@cebas.csic.es (E.M.-L.); bgosalvez@cebas.csic.es (B.G.B.); 2Abiopep S.L., R&D Department, Parque Científico de Murcia, Ctra. de Madrid, Km 388, Complejo de Espinardo, Edf. R, 2º, 30100 Murcia, Spain; cgaix@abiopep.com

**Keywords:** plant virus–host interactions, viral life cycle, viral replication, viral transmission, imaging, light microscopy, electron microscopy, correlative microscopy, in situ hybridization, immunocytochemistry

## Abstract

Plant viruses are obligate parasites that need to usurp plant cell metabolism in order to infect their hosts. Imaging techniques have been used for quite a long time to study plant virus–host interactions, making it possible to have major advances in the knowledge of plant virus infection cycles. The imaging techniques used to study plant–virus interactions have included light microscopy, confocal laser scanning microscopy, and scanning and transmission electron microscopies. Here, we review the use of these techniques in plant virology, illustrating recent advances in the area with examples from plant virus replication and virus plant-to-plant vertical transmission processes.

## 1. Introduction

The obligate parasitic nature of viruses, associated to the small repertoire of proteins encoded by their tiny genomes, makes tight and finely-tuned virus–host interactions indispensable for them. Understanding the mechanisms underlying these interactions may provide better knowledge of the viruses’ biology as well as the fundamental processes of plant physiology, and this may even point to novel means of controlling these harmful pathogens. The successful infection of a plant by a virus involves a few basic steps, which include entry, desencapsidation, translation, replication, cell-to-cell movement, encapsidation, vascular transport, and plant-to-plant transmission, which can be horizontal through vectors or mechanical wounds, and/or vertical through seeds and pollen. No less important is the fact that along these steps, viruses need to counter host defenses, including RNA silencing as well as protein-mediated basal and specific immunities. In the lines below, we briefly review the imaging techniques used to study plant–virus interactions during these steps. We then illustrate this first part with examples from the virus replication and vertical host-to-host transmission steps, which include processes that we have studied throughout our scientific careers as plant virologists.

## 2. Imaging Techniques Used in Plant Virology

### 2.1. Macroscopic Techniques

Before embarking in a microscopy study, it is fundamental to understand how the virus is distributed in the tissues under observation to precisely define the specimen to be placed under the microscope; for this, the use of a macroscopic technique such as hybridization on tissue-prints may provide optimal results. Tissue-printing hybridization is a simple technique for transferring macromolecules, such as viral nucleic acids, onto positively charged nylon or nitrocellulose membranes blotted directly from plant organ surfaces. The resulting blot is an image of the tissue surface. Initially, immobilized viral target nucleic acids were detected using the hybridization of radiolabeled nucleic acid probes and exposure to X-ray film, but later, digoxigenin-labeled nucleic acid probes combined with chemiluminescent detection became the norm [1] (Figure 1). Tissue-printing hybridization can be used as an initial approach to study the long distance movement of viruses in fully susceptible or partially resistant plants [2,3,4,5,6,7]. Moreover, this technique can be used to colocalize both host mRNAs and viroid or virus RNAs in plant tissues [8,9,10,11,12].

### 2.2. Light Microscopy

Plant tissue is quite difficult to prepare in order to obtain good specimen preservation. The cell wall and large vacuoles present in plant cells cause problems of osmolarity, producing trouble with fixation, dehydration, and embedding. It is important for both light and transmission electron microscopy studies to try different fixation methods, especially for protein or nucleic acid localization experiments. For example, we successfully used combinations of glutaraldehyde plus p-formaldehyde [4,13,14], although *p*-formaldehyde or formaldehyde alone could also give good results when working with plant tissues [3,15]. Choosing an adequate embedding medium is also important. For plant tissues, paraffin embedding is quite adequate for light microscopy [16], although it is time consuming. Alternatives to chemical fixation could be considered when the required equipment is available, including cryofixation and high pressure freezing [17].

Light microscopy is frequently complemented by the immunodetection of specific host or viral proteins. As with light microscopy, immunodetection can also complement electron microscopy, and both share similar methodological constraints. For light microscopy, specific antibodies are usually detected using a digoxigenin (DIG)-conjugate anti-immunoglobulin secondary antibody, whereas for electron microscopy, antibodies conjugated with gold particles are used (see in next Section 2.3). Likewise, in situ hybridization (ISH) provides the means to detect specific plant and viral nucleic acids in plant tissue sections. Similar to immunodetection, ISH can provide cellular resolution and in some cases, subcellular resolution. ISH has been widely used to provide spatial and temporal information on the localization of host and virus nucleic acids [2,4,5,6,13,14,18,19,20]. ISH involves the hybridization of tagged nucleic acid probes to target viral nucleic acids within thin tissue sections followed by the detection of the tagged probe with an anti-tag antibody [21]. The most commonly used probes are short (150–300 nucleotides), in vitro synthesized single-stranded complementary RNAs (cRNAs), or DNAs (cDNAs) labeled with digoxigenin-11-UTP or biotin. Hybridized tagged probes are detected using anti-tag antibodies conjugated to alkaline phosphatase (AP) or horseradish peroxidase (HRP). After incubation with AP- or HRP-appropriate substrates, a chromogenic signal reveals the precise histological distribution of the viral nucleic acid under the light microscope. By combining differentially-targeted probes with different tags, it is possible to colocalize the nucleic acids of different viruses in a tissue section. Alternatively, the observation of consecutive sections, each hybridized with a different probe, may allow for the colocalization of the nucleic acids of different viruses. We used this methodology to colocalize two virus isolates belonging to different strains in mixed infections (Figure 2) [16]. A number of variations to the traditional ISH protocols have been reported, including fluorescence in situ hybridization (FISH) [22,23], whole mount ISH (WISH) [14] and in situ PCR and RT-PCR [24,25,26,27,28,29,30,31], although PCR and RT-PCR have not been widely used in plant virology. WISH is an alternative for the localization of proteins or nucleic acids in small nonsectioned specimens [14] (see below in Section 4), but it is not always feasible.

### 2.3. Electron Microscopy

Electron microscopy is at the heart of plant virology, and it has been broadly used as the chief method to characterize plant tissues and cells infected by viruses and to describe the alterations induced in infected hosts [32,33,34,35,36,37,38,39,40,41]. Viral particles can be confounded with other particles and/or contaminants in virus preparations or in tissue sections; thus, for specific virus detection, electron microscopy coupled to immunolabeling is the preferred option. Immunogold labelling (IGL), where specific antibodies are linked to gold particles, is a powerful technique for the detection and localization of antigens in thin sections. Gold particles are highly electron dense and thus are able to show the position on a grid of any molecule to which they are attached. IGL has been successfully used for a wide variety of plant virus particles and proteins [42,43,44]. The gold particles are available in a wide range of sizes, from the smaller 1–5 nm ones, to the largest 20–25 nm in size, so choosing antibodies linked to different size particles can allow for detecting more than one antigen in a given tissue section. While IGL has been successfully used in a good number of experimental systems, problems attributable to the fixation and the embedding processes are still a source of recurrent trouble; these steps are crucial for preserving the ultrastructure of the cells as well as immunogenic epitopes. It is thus important, although not always possible, to try different fixation methods, for instance, fixation in glutaraldehyde alone versus glutaraldehyde followed by osmium tetroxide, as well as combinations of glutaraldehyde plus p-formaldehyde. In general terms, glutaraldehyde is better for preserving the cell and tissue structure, while p-formaldehyde provides better antigenic preservation. Similarly, it is important to use appropriate embedding resins such as LR-White or LR-Gold, or low temperature polymerization resins such Lowicryl. Another source of problems may be the specificity of the labelling antibody. In this case, cross absorption of the antiserum with a protein extract from healthy host tissues may result in a significant decrease of the nonspecific signal background.

Specific antibodies can also be used in combination with electron microscopy for the fast and accurate detection of viral particles. Immunosorbent electron microscopy (ISEM) consists of decorating glow discharge-treated TEM grids directly with a coating antibody or with protein A that binds the antibody and immune-immobilized viral particles from a crude tissue sap [45,46,47,48,49]. ISEM seems to be particularly useful in encapsidation/decapsidation studies, where viral particles may be scant and an enrichment process is required prior to observation [50,51].

### 2.4. Fluorescence Microscopy

Fluorescence methods, including confocal techniques, are generally used to track specific molecules tagged with fluorescent labels within cells, to localize them in organelles and/or to colocalize them with other target molecules [52]. Fluorescent microscopy imaging can be performed using wide-field (or “regular”) epifluorescence or confocal laser scanning microscopy (CLSM). Wide-field microscopy requires cheaper equipment than CLSM and can provide high-quality images; however, out of focus fluorescence captured during wide-field microscopy can significantly reduce the signal-to-noise ratio in the image, and therefore CLSM is generally preferred. Fluorescence microscopy became popular after the discovery and engineering of infrared fluorescent proteins (IFPs). The original green fluorescent protein (GFP) was isolated from the jellyfish *Aequorea victoria*; it has major excitation peaks at 395 and 470 nm and emits green fluorescence at 520 nm [53]. Mutant screenings and screenings of other fluorescent marine organisms have led to the identification of other IFPs, including the red emitting DsRed or mCherry, and thus there is now a plethora of IFPs with varied excitation and emission characteristics [54,55]. It is important to consider that plant tissues contain autofluorescent compounds with broad excitation and emission spectra, and although the signal coming from them is generally weak, it can bleed through into the IFPs channels. Chlorophyll is often the most problematic autofluorescent compound, and its signal can be detected in the red IFPs’ channel. If crosstalk between chlorophyll and red IFPs is anticipated, it is advisable to set up optimal observation parameters with untreated samples. Crosstalk can also be of concern when simultaneous localization of two or more IFPs is intended. Similar emission intensities and sufficiently differing wavelengths between IFPs are required, and sequential image acquisition of the different channels is recommended [56].

The availability of infectious virus clones has enabled viral genome engineering to introducing IFP-expressing genes, which allow for the localization of viral gene products, commonly in the form of fusion proteins [57,58] (Figure 3). IFPs expressed from the viral genome as fusion proteins have been used to track the intracellular localization of viral proteins to study different aspects of the infection process. Different methods that allow for more detailed investigation of different features have been devised. For instance, to study in vivo protein–protein interactions, bimolecular fluorescence complementation (BIFC) is very frequently used [59,60,61,62,63], or to discover when the protein is active, fluorescence resonance energy transfer (FRET) and fluorescence lifetime imaging (FLIM) have been used [64,65,66,67,68]. Other techniques have been useful for the study of the structure of biological membranes and to measure the movement (e.g., diffusion) of proteins in a living cell, such as fluorescence recovery after photobleaching (FRAP) [69,70,71]. In addition, trafficking between organelles has been studied with fluorescent microscopy techniques such as fluorescence loss in photobleaching (FLIP) in order to study ER-Golgi trafficking or the continuity of intracellular organelles [72], among others. These methods are not reviewed here due to space constraints, but we refer the reader to the corresponding citations.

### 2.5. Three-Dimensional (3D) Imaging Techniques

Transmission electron microscopy (TEM) provides only random or discontinuous pictures of cellular organelles [73] and thus may lead to misconceptions regarding cellular ultrastructure. For example, a three-dimensional (3D) analysis of turnip mosaic virus (TuMV)-induced intracellular rearrangements revealed that the vesicle-like structures in two-dimensional TEM images were, in fact, tubules [17]. To overcome the limitations of the random sectioning used in traditional TEM analysis, many novel 3D TEM methods have been developed, including serial sectioning, electron tomography (ET), scanning transmission electron microscopy (STEM) tomography, serial block-face SEM (SBF/SEM), focused ion beam/field emission scanning electron microscopy (FIB-FESEM), cryo-ET, and cryo-FIBSEM [74]. Cryofixation and ET are probably the most prominent technical advances in TEM since the beginning of the millennium [75,76]. Although not applicable to many objects of study, ET of cryofixed samples in vitreous ice (cryo-ET) has revealed organelle structures at subnanometer scales [77]. High-pressure freezing (HPF), followed by freeze substitution (FS), is a more practical cryofixation protocol for plant cell samples and has been widely used to process specimens for ET [78]. Other powerful 3D imaging technologies, such as SBF/SEM and FIB-FESEM, have the additional advantage of making possible the reconstruction of large volume structures, and they can be used to overcome the thickness limitations of ET. With SBF/SEM and FIB/FESEM, sectioning is performed automatically inside the SEM microscope using a diamond knife or a focused ion beam [74]. The main advantages of FIB/FESEM over SBF/SEM are the significant improvement in the Z-axis resolution and the possibility of targeting a small region of interest without destroying the remainder of the block face [79,80]. Hence, FIB/FESEM has been successfully used, although in very few occasions, for the study of structures in virus infected cells [81,82,83].

### 2.6. Correlative Light Electron Microscopy (CLEM)

Correlative microscopy consists of the use of two or more microscopy techniques to characterize the same region of interest in a sample [84]. Correlative microscopy has been made possible due to the availability of advanced sample preparation and modern labelling approaches that can highlight the structures of interest on multiple microscopy platforms [85,86]. One of the major strengths of correlative microscopy is its ability to use specific labels or probes that localize structures of interest during light or fluorescent microscopy, after which electron microscopy is used [87]. Over the past decade, CLEM has been applied to a range of plant tissues, including leaves [88,89,90], stems [91,92], reproductive organs [93], and roots [94], and it was used as well to study the cell cytoskeleton [95]. Despite being used in much research, CLEM has not been widely used for plant virus studies.

New techniques such as super-resolution microscopy have already been used, for instance, for the study of the viral factories induced by potato virus X (PVX) [96], making it possible to have major advances in this area of research (see below in Section 3.3). However, further work is necessary to transfer the latest technological advances, such as the development of specific tags for electron microscopy [97], and the use of CLEM, to increase our current knowledge of the biological processes underlying plant virus–host interactions.

## 3. Use of Imaging Techniques to Study Plant Virus Replication

Replication is the central step of the virus infection cycle and aims at producing the viral progeny. As the majority of plant viruses have single stranded (ss) positive sense (+) RNA genomes [98], the examples in the following sections refer to this type of viruses.

### 3.1. Genesis of Plant ss(+)RNA Virus Replication Factories

Viral RNAs (vRNAs) associate with viral proteins to form viral ribonucleoprotein complexes (vRNPs) of variable nature. vRNPs are the key players of the viral infectious cycle, driving various processes such as translation, transcription, and intracellular trafficking. The replication process of ss (+) RNA viruses is carried out in virus-specific replication complexes (VRCs) formed by the RNA-dependent RNA polymerase (RdRp; viral replicase), the replication-auxiliary proteins from the virus, vRNAs, and factors from the host, which are assembled together in association to certain organelles [99]. VRCs include extensive host endomembrane reorganizations to create a favorable microenvironment for virus replication [97,98,100,101]. The recruited organelles for building the replication factories depend on the virus. The proliferation and modification of the endoplasmic reticulum (ER) membranes have been described in plants infected by members of the genera *Potyvirus* [41,102,103], *Nepovirus* and *Comovirus* [46,104,105], *Potexvirus* [106], *Tobamovirus* [107,108] and *Bromovirus* [109]. Turnip yellow mosaic virus (genus *Tymovirus*), barley stripe mosaic virus (genus *Hordeivirus*) and bamboo mosaic virus (genus *Potexvirus*) use chloroplasts to build their replication sites, while members of the family *Tombusviridae* use the ER, peroxisomes, or mitochondria (see below in next section).

Membrane alterations leading to the production of replication factories depend on the action of one or more viral proteins, with viral replicases being the major contributors to the endomembrane alterations that occur during virus infection [110,111,112]. These proteins either are integral membrane proteins with localization signals that target a given organelle or interact with membrane proteins or lipids to drive their specific localization. The expression of these proteins alone may induce the formation of altered membranous structures similar to those found in virus-infected cells [113], although the remodeled membranes may differ from those observed during infection [114]. Replication proteins also recruit other viral and host proteins or RNAs to the replication sites, leading to membrane bending and deformation. Two morphotypes of replication factories that share basic morphological similarities seem to exist for ss(+)RNA viruses; one morphotype is characterized by the formation of invaginated spherules with neck-like channels that connect the interior of the spherule to the cytoplasm, and the other morphotype is characterized by the presence of single and/or double-membrane vesicles that are formed by endomembrane remodeling [110,115,116]. Examples for plant viruses are presented in the below subsections. Despite recent advances, our current knowledge of the 3D architecture of ss(+)RNA virus replication factories is limited to a few experimental systems and is largely descriptive; thus, there are still many issues that need to be addressed.

### 3.2. Replication Factories of Viruses of The Family Tombusviridae

The family *Tombusviridae* comprises sixteen genera including tombusviruses, betanecroviruses, and alfa-, beta-, and gamma-carmoviruses. The examples below suggest that viruses in this family tend to build VRCs of the morphotype characterized by the formation of invaginated spherules with neck-like channels, although they differ in the replication organelle used.

Tombusviruses have a wide host range and are among the best characterized plant viruses [117,118,119,120]. Their genome is ~4.8 kb in size and encodes five proteins, including two essential replication proteins, p33 and p92pol, with the latter being the RdRp that is translated from the genomic RNA via the read through of the translational stop codon in the p33 ORF [121]. For tomato bushy stunt virus (TBSV), a type member of the genus *Tombusvirus*, replication takes place in interconnected single-membrane vesicle-like structures 80–150 nm in diameter formed in peroxisomal boundary membranes induced during infection [122,123,124,125]. Structures identical to these are formed in yeast transformed with a TBSV-based replicon that expresses p33 and p92pol. Metal-tagging TEM and 3D molecular mapping were used to reconstruct the TBSV replication factories in yeast, showing that they consist of large numbers of spherules that are connected through a neck to the peroxisomal multivesicular body boundary membrane [126]. In the yeast strain *PAH1-*, which is defective in peroxisomal formation, replication takes place efficiently [127,128], but replication factories are constituted by spherules that are connected to expanded ER membrane stacks that create a network. These ER structures may originally correspond to nascent peroxisomes, as the biogenesis of peroxisomes involves ER membranes [126].

TBSV and the closely related cucumber necrosis virus (CNV; genus *Tombusvirus*) use peroxisomal membranes, whereas beet black scorch virus (BBSV; genus *Betanecrovirus*) builds its replication factories in vesicles packets along the ER or in the perinuclear cytoplasmic region [114]. ET has been used to build a 3D model where one to several hundreds of spherules 50–70 nm in diameter are found within the vesicles packets and contain BBSV double-stranded RNA (dsRNA) and the replication protein p23, suggesting that these spherules are sites of viral replication, as dsRNA is a replicative intermediary for ss(+)RNA viruses. Most BBSV spherules have a narrow neck connecting the spherule interior to the cytoplasm, suggesting that they are formed from the invagination of ER membranes [114]. Interestingly, the vesicles packets induced by BBSV appear to be connected to each other by tubule-like structures that are 15–30 nm in diameter [114].

Viruses in the three carmovirus genera seem to share the use of mitochondrial membranes to build their replication factories. Pelargonium flower break virus (PFBV; genus *Alphacarmovirus*) induces enlargements of the mitochondrial cristae which often contain fibrillar material [129], and extensive peripheral vesiculation of mitochondria was observed in turnip crinkle virus (TCV; genus *Betacarmovirus*) infected turnip cells [130,131]. For melon necrotic spot virus (MNSV; genus *Gammacarmovirus*), we have shown that viral infection induces deep modifications of the mitochondrial ultrastructure, including dilated cristae, a clear separation of the double membrane of the mitochondrial matrix, and large dilations which can occupy up to 80% of the mitochondrial interior, with numerous vesicles that are 45–50 nm in length along the external membrane of the mitochondria, and in the internal lumen of the dilations. Vesicles were often bottle-shaped, opening either towards the cytoplasm or to the dilation lumen (Figure 4). The localization of viral CP, ss(+)RNA, and dsRNA indicated that altered mitochondria were in fact the MNSV replication sites [83]. FIB/FESEM and posterior 3D image reconstruction showed that altered mitochondria contain large internal interconnected dilations linked to the external cytoplasmic environment through pores and with lipid bodies (Figure 5) [83]. By transiently expressing the p29 MNSV replication auxiliary protein, we revealed the specific targeting of this protein to mitochondria. In fact, the small replicase protein of tombusviruses appears to be an integral membrane protein that usually contains sequences that target a particular cell organelle in which it associates with membranes, inducing drastic modifications in them [118,132,133,134,135,136]. However, the expression of this protein may not be sufficient for the formation of specific vesicles, suggesting that other factors are probably involved in the induction of these structures. For example, the cellular protein ESCRT, involved in tombusvirus-induced membrane deformations, is required for the assembly of the viral replicase [125]. The MNSV p29 has three transmembrane domains (TMD). According to Mochizuki et al. [137], the presence of TMD2 is crucial for its mitochondrial localization, where it may be anchored to the membrane, inducing degradation of the organelle and, hence, cell necrosis [137]. Our work [83] with the ectopic expression of p29 fused to GFP showed that this protein is targeted to mitochondria, where, unlike carnation Italian ringspot virus-induced modifications [134], it not only induces its disorganization but promotes the proliferation of mitochondrial membranes in a very specific manner, either by the induction of double membranes, ending in complete disruption of the organelle, or by the formation of double membranes surrounded by dilations that result in very similar structurally altered mitochondria [83]. Since p29 and p89 (the MNSV RdRp) share the first two TMD, it would be reasonable to propose that both are directed toward the same organelle, inducing vesicle formation with the likely presence of both viral proteins and even RNA inside.

The manner in which replication and movement are orchestrated is still poorly understood. It was shown that a protein complex, named endoplasmic reticulum (ER)-mitochondria encounter structure (ERMES), is located on the interface of the ER and the mitochondria, providing a hub for fine communication between both organelles [138]. Although ERMES was not involved in PFBV p27 targeting to mitochondria in yeast [132], it is tempting to speculate that MNSV could take advantage of this complex to interconnect replication and movement. Interestingly, we reported on a frequent association of the MNSV viral factories to the ER in close proximity to the plasmodesmata (Pd) (Figure 4C) [83], therefore suggesting the existence of intercellular transport mechanisms that function more directly, in a similar way to those described for potato virus X (PVX) where viral RNA is coreplicationally delivered to Pd (see next section).

### 3.3. Potexvirus Replication Factories

The genus *Potexvirus* belongs to the family *Alphaflexiviridae*. The potexvirus genome is an ss(+)RNA molecule with five open reading frames (ORFs). ORF1 encodes the viral RNA-dependent RNA polymerase (RdRp), which is the only viral protein that is absolutely required for viral replication [39,139]. ORFs 2 to 4 comprise the triple gene block (TGB) module encoding the TGB1, TGB2, and TGB3 proteins, respectively. ORF5 encodes the viral coat protein (CP). PVX is the potexvirus type member, and the PVX VRC is the best studied potexvirus replication factory. During infection, PVX induces the formation of large perinuclear aggregates, which were initially named X-bodies. The TGB1 protein reorganizes actin microfilaments in aggregates, where ER and Golgi vesicles are recruited by proteins TGB2 and TGB3 to form the X-body [39]. X-bodies contain the five viral proteins, vRNA, and ribosomes [38,39,40,43,106]. CLSM, 3D structured illumination (3D-SIM) super-resolution microscopy, and IGL in TEM have shown that the X-bodies have a defined stratified structure: TGB1 aggregates in the core of the X-body surrounded by actin microfilaments; in the next layer, there are ER and Golgi-derived vesicles associated with TGB2 and TGB3 proteins colocalizing with naked vRNA; finally, virions surround the overall body in a cage-like fashion [39,96]. It was proposed that viral RNA replication takes place in the layer between TGB1 aggregates and TGB2/3 granules, as abundant nonencapsidated vRNAs were observed in this area [39]. During early infection, PVX also induces cap-like complexes at the entrances of Pd that have a composition and structure that are similar to those of perinuclear X-bodies [40], suggesting that the cap-like complexes are small virus factories that couple viral protein translation, RNA replication, particle encapsidation, and intercellular movement [39,40]. Recently, Wu et al. [140] analyzed the localization of dsRNA and PVX RdRp, TGB2, and TGB3 in living *Nicotiana benthamiana* cells. Their results revealed that TGB2 and RdRp colocalize with dsRNA in small motile cytoplasmic aggregates that appear in the early stage of infection, in the cap-like complexes, and in the X-bodies. These authors also proposed a model in which the motile foci in the cytoplasm slowly cluster into the large, irregularly-shaped aggregates in the perinuclear area constituting the X-body, with the entire process orchestrated by the interaction of the C-terminal domain of the PVX RdRp and the TGB2 [140].

The results we have obtained from studying pepino mosaic virus (PepMV) (genus *Potexvirus*, family *Alphaflexiviridae*) infections of *N. benthamiana* and tomato plants seem to conform to the model proposed for PVX. The PepMV genome has an identical organization and encodes proteins homologous to those of PVX [141]. Minicka et al. [36] carried out ultrastructural and immunocytochemical analyses of tomato tissues infected with different PepMV isolates, but no clear evidence was obtained on the possible nature and localization of the PepMV VRCs, even though different cytopathological alterations were observed. We developed PepMV-based vectors expressing fluorescent proteins [57,58], which are useful for cell biology studies. In using them and CLSM, we observed cytoplasmic aggregates induced by the virus, normally in the vicinity of the cell nucleus and only one per cell, which likely constitute the PepMV VRCs. We also observed that the ER and Golgi apparatus were reorganized in the infected cells and recruited to the PepMV VRCs (Figure 6) [58].

### 3.4. Potyvirus Replication Factories

The genus *Potyvirus* within the family *Potyviridae* comprises a very large number of species including some harmful plant pathogens. The potyvirus genome consists of a ss(+)RNA molecule of approximately 10 Kb in length, which encodes a single polyprotein that is processed by proteinases into several functional proteins; it also encodes a small fusion protein produced by a +2 frameshift. Potyvirus VRCs are ER-derived, membrane-bound complexes, and they appear to be primarily induced by the small membrane-associated 6K2 potyviral protein [142]. Potyviral translation occurs first at ER-derived membranes; after initial translation, the viral protein 6K2 remodels the ER membrane to recruit the vRNA and replication-associated proteins, resulting in the formation of the VRC [102,143,144]. The 6K2-containing VRCs exit the ER site and fuse with chloroplast membranes, where active viral replication takes place [145]. VRCs contain potyviral proteins and also host proteins, including the eukaryotic translation initiation factor 4E (eIF4E), eukaryotic elongation factor 1A (eEF1A), RNA helicase-like protein RH8, poly(A) binding protein (PABP), and heat shock protein 70 (HSP70) [143,146,147,148,149]. Turnip mosaic virus (TuMV) is a potyvirus that has been extensively used as an experimental model to characterize potyviral VRCs [111]. TuMV reorganizes the endomembrane system of the infected cell to generate endoplasmic reticulum-derived motile vesicles containing viral replication complexes; as for other potyviruses, the TuMV 6K2 protein plays a key role in the formation of these vesicles. Once vesicles are formed from the ER as convolutional membrane structures, they bud off from the ER exit sites and bypass the Golgi apparatus to follow different fates, maturing into replication competent single membrane vesicles [17], fusing with chloroplasts for efficient replication [150], or maturing into larger double-membrane vesicles and multivesicular bodies for cell-to-cell movement to continue with the infection process [17]. It also appears that vesicles can drive viral products to the extracellular space. Thus, the authors in [151] observed that the TuMV 6K2 protein can be localized in the extracellular space of infected *Nicotiana benthamiana* leaves. Using TEM, these authors observed the proliferation of multivesicular bodies during infection and their fusion with the plasma membrane that resulted in the release of their intraluminal vesicles to the extracellular space. Immunogold labeling showed that the released vesicles contained dsRNA of a likely viral origin. Focused ion beam-extreme high-resolution scanning electron microscopy was used to generate a 3D model that showed extracellular vesicles in the cell wall, and proteomic analyses confirmed the presence of viral proteins in the extracellular space [151]. This is a very interesting set of observations, but its biological meaning remains to be elucidated.

Recently, a potyvirus-induced granule (PG) structure was discovered in potato virus A (PVA) infected cells [152]. PGs are intermediate structures between PVA replication and translation. PGs are loosely associated with VRCs and are induced by the potyviral protein HC-Pro [152]. The TuMV VPg was recently found to resist the autophagy-mediated degradation of TuMV HC-Pro-induced RNA granules; thus, the emerging picture suggests that HC-Pro induces PGs to store vRNAs and protect them from host-mediated silencing until potyviral VPg can assist them in their translation [153].

### 3.5. Spatial Distribution of Active Replication Sites in Tissues and Whole Plants

With a few exceptions, the results described above were obtained after the analysis of infected cell patches that were directly inoculated; therefore, the cells in the inoculated patch can be considered synchronously infected at least up to a certain point. However, the infection of an entire plant is very far from being a synchronous process, with equivalent infection events taking place in just a small number of the plant cells. This has been nicely demonstrated in the early studies from Maule’s group, who showed that infections proceeded in plant tissues in waves. Active replication takes place only at the infection front, and this is concomitant with the up-regulation of several stress-responsive host genes and an apparently generalized host gene shutoff [18,19,154,155]. There are few studies dealing with the in planta accumulation kinetics of vRNAs and their translation products. In an example from our group, we used ISH to analyze the spatiotemporal accumulation of the RNAs, CP, and MP (p7) proteins of carnation mottle virus (CarMV; genus *Alphacarmovirus*, family *Tombusviridae*) in *Chenopodium quinoa* plants [2]. Viral (+) RNA accumulated in each leaf cell type (epidermis, palisade parenchyma, mesophyll, and vascular bundles) within the symptomatic areas [2]. However, unlike (+) RNA [2], viral (−) RNA was localized only on the edges of the chlorotic lesions [2]. Moreover, the signal on the leading edges on both sides of the lesion was connected throughout the central, noninfected area by infected spongy parenchyma and lower epidermis cells, suggesting a progression of the virus from the upper to the lower epidermis. Therefore, virus replication seemed to be spatially regulated, as it was localized behind the infection front or on the leading edges of the chlorotic lesions [2].

## 4. Use of Imaging Techniques to Study Vertical Transmission of Plant Viruses

Unlike animal viruses, where hosts are mobile and often come into contact with each other, plant viruses need to cover the distances that separate their immobile hosts. Many plant viruses are horizontally transmitted by biological vectors that feed on the plant. Vectored transmission is a vibrant area of research in plant virology, and many original studies and reviews have been published recently. These have covered different facets of this discipline, including cellular and molecular aspects of the plant virus–vector interactions analyzed by using imaging techniques [156,157,158,159]. In contrast, vertical plant-to-plant transmission has been less often studied using imaging techniques, in spite of the importance that vertical transmission has in the long-distance dispersal of plant viruses. Vertical transmission does not rely on vectors and occur directly from parents to their offspring, either via seeds (or other propagules) or pollen. Seed and pollen transmission are common spread mechanisms for approximately 20% of the plant viruses [160,161], with vertical transmission through seeds being important in about 15% of the plant virus species [160]. Tobamoviruses contaminate the seed coat externally and are later transmitted mechanically to the germinating plants [162]. However, “true” seed infection occurs through the embryo, via two distinct but sometimes coexisting pathways. On the one hand, the embryo can start its development infected if at least one of the two gametes that take part in the fecundation process has previously been infected. On the other hand, some plant viruses are capable of infecting the embryo after fecundation [163]. Virus transmission by pollen has been reported for more than 45 virus species [160,164]. However, only a few studies have shown how viruses infect male or/and female gametes through imaging techniques. For instance, an examination of barley flower tissues via ultrastructural studies with TEM revealed that the seed- and pollen-transmitted isolate MI-1 of the barley stripe mosaic virus (BSMV) invaded the floral primary meristem early and, subsequently, the pollen mother cells and sperm, as well as the megaspore mother cells including the egg [32,33]. Studies in pea infected with pea early browning virus (PEBV) showed that this virus was present in the synergids of the egg cells and the polar nuclei from unfertilized ovules, as well as in the mature pollen from unopened flower anthers [165]. Through immunolabelling and ISH in LM of developing anthers from young lettuce flowers infected by lettuce mosaic virus (LMV), Hunter and Bouyer [34] showed that the detection of LMV in pollen mother cells and the surrounding tapetum was consistently correlated. In an example from our group, prunus necrotic ringspot virus (PNRSV) was shown to invade early pollen grains, infecting the megaspore and generative cell of the bicellular pollen grain in nectarine [166] and apricot trees [14]. In apricot triporate pollen grains, we localized PNRSV via whole mount ISH in LM mainly at the apertures from which the pollen tube emerges. After pollen tube elongation, vRNA was localized on its end, in the growth zone (Figure 7) [14]. Thus, early positioning of PNRSV at the tube emergence apertures and later at the growing tips could increase the transmission opportunities of this virus during fertilization [6].

Direct colonization of the embryo has been described as an alternative entry mode for BSMV, and it represents the only mechanism for pea seed borne mosaic virus (PSbMV) transmission in pea [35,167]. Direct infection of the embryo occurs by virus entry into the seed from the surrounding maternal testa, and subsequent movement to the endosperm, suspensor, and finally, the embryo. This movement must occur symplastically during early embryogenesis while the suspensor is still functional, i.e., prior to its programmed cell death (apoptosis), representing a narrow temporal window for embryo infection [167]. Embryo infection does not necessarily result in seed transmission, i.e., the infection of seedlings from infected seeds. For instance, PNRSV infects all seed parts, including the embryo at early developmental stages [6]. However, the germination of seeds collected from PNRSV-infected apricot trees showed that this virus was seed-transmitted in only 10% of infected seeds, suggesting that virus inactivation takes place during seed maturation [6,20].

## 5. Conclusions and Future Trends

Advanced imaging techniques, such as ET, have a great potential for the further understanding of the cellular context in which molecular virus–host interactions take place. In this regard, CLEM has scarcely been used in plant virology, if ever, but we are convinced that it has enormous possibilities. CLEM can be better performed by preserving the specimen for EM observation through high-pressure freezing and cryofixation. Access to the sophisticated equipment and the trained personnel required for this is a limiting factor for performing CLEM; hopefully in the future, a more generalized use of CLEM will make this technique more accessible to plant virology studies.

Genetic, biochemical, and molecular biology approaches are contributing to the identification of host factors required to complete the viral cycle. Complementing these approaches with cellular biology and imaging techniques is critically helping us to gain a better understanding of the roles played by host factors both for the cell and for the virus, and together they can provide a very detailed picture of the molecular mechanisms underlying the multiple processes taking place during the viral cycle. The more the number of added experimental systems are, the more refined the picture will be, and this includes not only different viruses, but also hosts that are different to *N. benthamiana* and *Arabidopsis*, i.e., the plant models that are traditionally used in plant cellular biology.

## Figures and Tables

**Figure 1 viruses-13-00358-f001:**
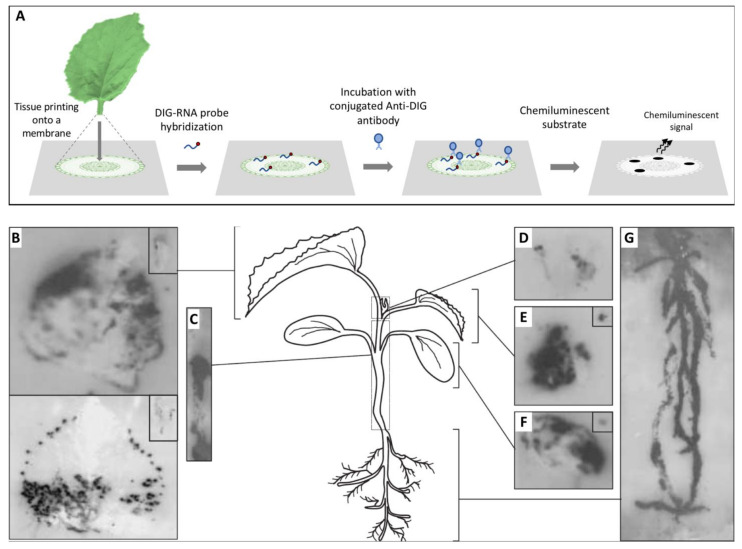
Analysis of the spatial distribution of a virus in infected plants by tissue printing hybridization. (**A**) Schematic representation of the tissue printing hybridization technique. Frequently after sectioning, the organ surface is printed onto a positively charged nylon or nitrocellulose membrane. Macromolecules, including viral nucleic acids, are transferred from the tissue to the membrane. The blotted membrane is then incubated with a digoxigenin (DIG)-labeled nucleic acid probe and the target nucleic acid is detected after incubation with a conjugated anti-DIG antibody and the appropriated chemiluminescent substrate. (**B**–**G**) Tissue print hybridization using a digoxigenin-labelled RNA probe for detecting melon necrotic spot virus (MNSV) vRNA in (**B**) the first systemic leaves (developed leaves), (**C**) hypocotyl and main stem, (**D**) shoot tip, (**E**) second systemic leaf (young leaf), (**F**) inoculated cotyledon, and (**G**) roots of MNSV-infected melon plants. Insets displayed in B, E, and F corresponds to longitudinal or cross-sectional printings of the petioles. This figure is adapted with permission from [8].

**Figure 2 viruses-13-00358-f002:**
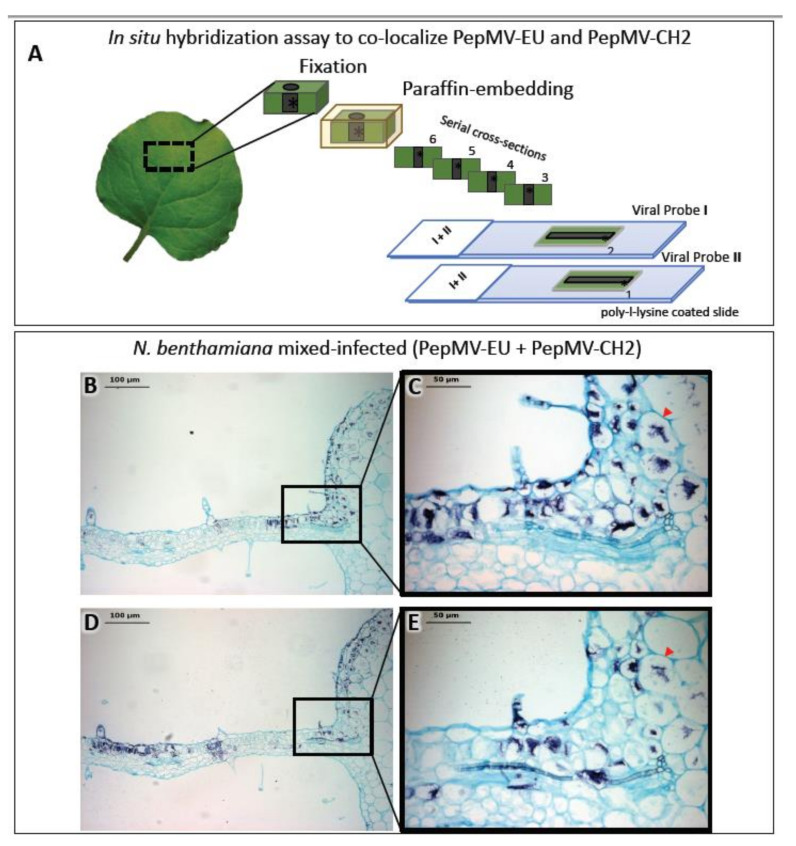
In situ hybridization (ISH) assay on consecutive serial cross sections of *Nicotiana benthamiana* leaves infected with two isolates belonging to different strains of pepino mosaic virus (PepMV-EU and PepMV-CH2). (**A**) Schematic representation of the ISH assay. Small pieces of mock and mixed infected *N. benthamiana* leaves were fixed with paraformaldehyde (PFA), embedded into paraffin and sectioned. The ISH was performed on consecutive leaf cross sections of the same sample by using each probe (I or II) in each slide (1 or 2), respectively. (**B**,**D**) Images of the ISH performed on a pair of consecutive leaf serial cross sections from mixed (PepMV-EU + PepMV-CH2) inoculated plants with either the riboprobe for PepMV-EU (**B**) or PepMV-CH2; both viruses are located in dark-blue colored areas of the leaf sections distributed in patches of infected tissue, mixed with areas of noninfected tissue. (**C**,**E**) Higher magnification of the area boxed in **B**, **D**, respectively. The viral RNAs of both PepMV-EU and PepMV-CH2 isolates are located in the same dark-blue colored cells (arrowhead) of the leaf, mixed together with other noninfected tissue cells, or only infected by one of the isolates. Scale bars are displayed in the images. This figure is adapted with permission from [16].

**Figure 3 viruses-13-00358-f003:**
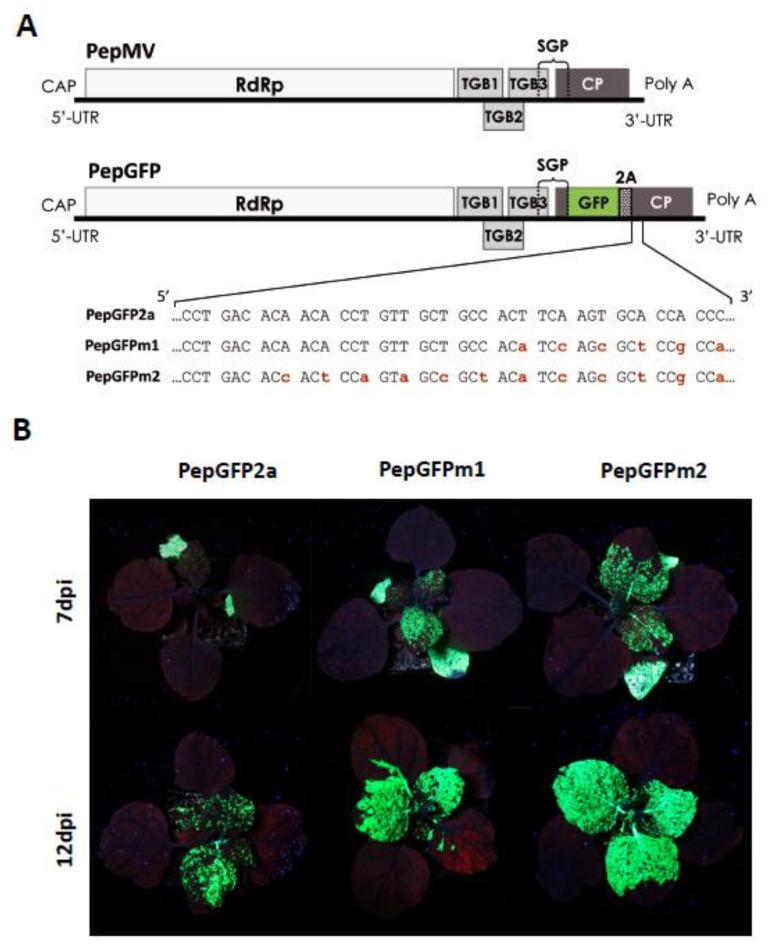
Pepino mosaic virus (PepMV) vectors expressing the green fluorescent protein (GFP). (**A**) Schematic representation (not to scale) of the PepMV genome and modified variants PepGFP2a, PepGFPm1, PepGFPm2, carrying the GFP gene. The GFP was expressed as a fusion to the coat protein (CP) through the foot and mouth disease virus 2A catalytic peptide sequence. The nucleotides marked in red correspond to synonymous mutations introduced into the corresponding vector versions to avoid sequence duplications. (**B**) *Nicotiana benthamiana* plants infected with PepMV vectors expressing GFP. Fluorescence was visualized in plants inoculated with PepGFP2a, PepGFPm1 or PepGFPm2 under UV light at 7 and 12 d post-inoculation (dpi). This figure is adapted with permission from [58].

**Figure 4 viruses-13-00358-f004:**
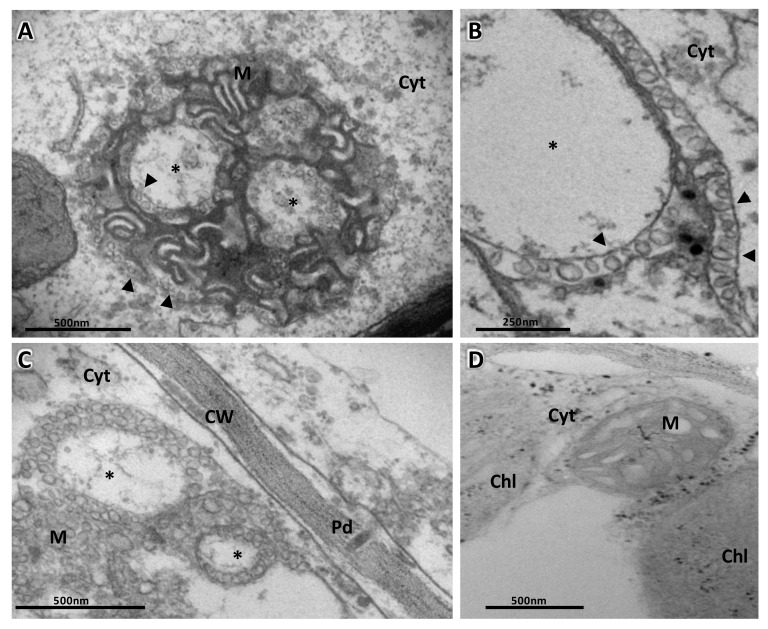
Transmission electron microscopy study of melon necrotic spot virus (MNSV)-infected cells from melon cotyledons. (**A**) Altered mitochondria (M) where the replication of the virus takes place, showing the presence of small vesicles (black arrow) inside large dilations (star) and around the periphery of the organelle. (**B**) Detail of a big dilation surrounded by bottlenecked vesicles connecting either with the dilation lumen or with the cytoplasm. (**C**) MNSV-altered mitochondria in the proximity of plasmodesmata. (**D**) Ultrastructure of a healthy plant mitochondria. Samples were taken 3 days post-inoculation. Notes: Chl, chloroplast; CW, cell wall; Cyt, cytoplasm; M, mitochondria; Pd, plasmodesmata. Scale bars are displayed with the images.

**Figure 5 viruses-13-00358-f005:**
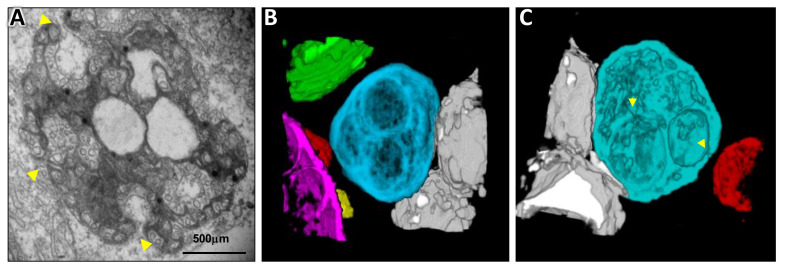
Transmission electron microscopy (TEM) analysis and 3D reconstruction of melon necrotic spot virus (MNSV)-altered mitochondria by focused ion beam/field emission scanning electron microscopy (FIB-FESEM). (**A**) TEM image of the samples used for FIB/FESEM showing the altered mitochondria ultrastructure with big dilations inside, and numerous vesicles in the external membrane and inside the organelle (arrowheads). (**B**) 3D model of the complete altered mitochondria (blue) next to other organelles such as chloroplasts (green), lipid bodies (grey) and other modified mitochondria (red, yellow and purple). (**C**) 3D reconstruction of the infected mitochondria using a partial series of FIB/FESEM images showing the presence of pores connecting the different internal dilations, as well as the lumen of the dilations to the cytoplasm. The connection pores are indicated with yellow arrowheads. This figure is adapted with permission from [83].

**Figure 6 viruses-13-00358-f006:**
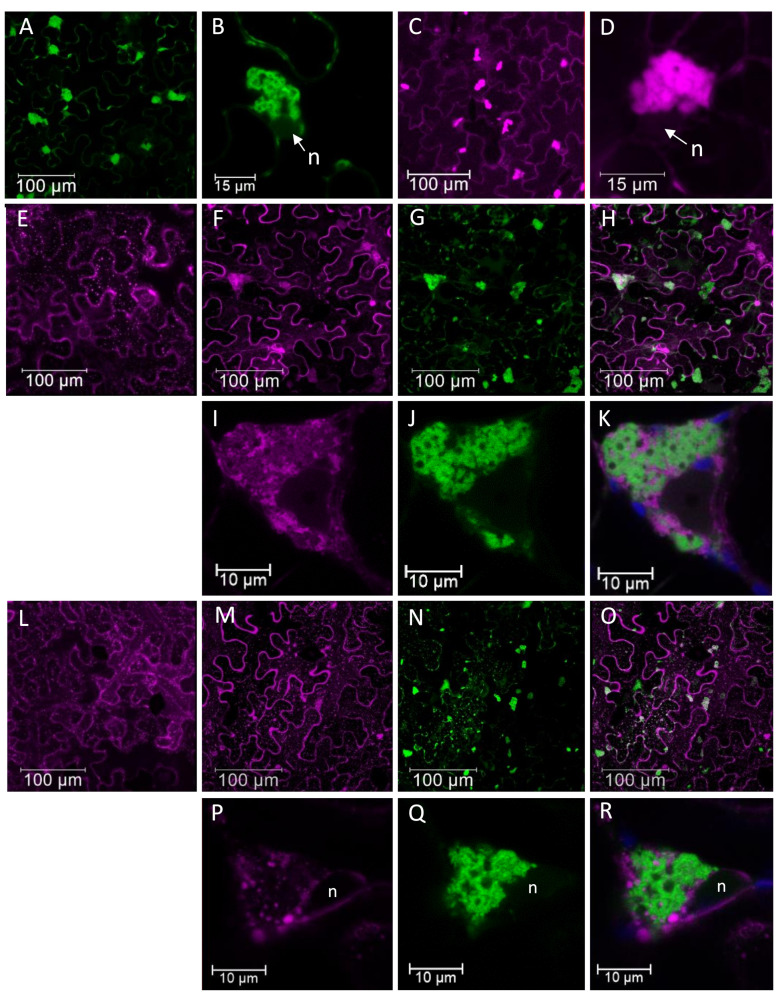
Pepino mosaic virus (PepMV)-induced subcellular bodies in *Nicotiana benthamiana* and tomato plants. (**A**) and (**B**) confocal laser scanning microscopy (CLSM) images of PepGFPm2 infection in *N. benthamiana* at 3 days post-inoculation (dpi). (**B**) High magnification of a fluorescent body in PepGFPm2 infection to show its cytoplasmic localization and spatial relation with the nucleus. (**C**) and (**D**) CLSM images of PepDsRed infection in tomato at 3 dpi. (**D**) High magnification image of a fluorescent body in PepDsRed infection. (**E**) Distribution of the endoplasmic reticulum marker (ER-mCherry) at 3 dpi in *N. benthamiana* epidermal cells in the absence of PepMV infection. In the presence of PepGFPm2 infection: (**F**) changes of ER-mCherry localization were observed, (**G**) green fluorescent bodies of PepGFPm2 and (**H**) merged image of (**F**) and (**G**) to see the matching. High magnification images: (**I**) the distribution of ER-mCherry during the infection, (**J**) the PepMV subcellular body and (**K**) merging of (**I**) and (**J**) to show the incomplete matching between the red and the green labelling in the body. (**L**) Distribution of Golgi-mCherry marker at 3 dpi in *N. benthamiana* epidermal cells in the absence of PepMV infection. In the presence of PepGFPm2 infection: (**M**) changes of Golgi-mCherry localization were observed (**N**), green fluorescent bodies of PepGFPm2 and (**O**) merged image of (**M**) and (**N**) to show the matching. High magnification images: (**P**) the distribution of Golgi-mCherry during the infection, (**Q**) the PepMV subcellular body, and (**R**) merged image of (**P**) and (**Q**) to show the incomplete matching between the red and the green labelling in the body. Notes: n, nucleus. Blue color corresponds to chloroplasts autofluorescence. Scale bars are displayed with the images. This figure is adapted with permission from [58].

**Figure 7 viruses-13-00358-f007:**
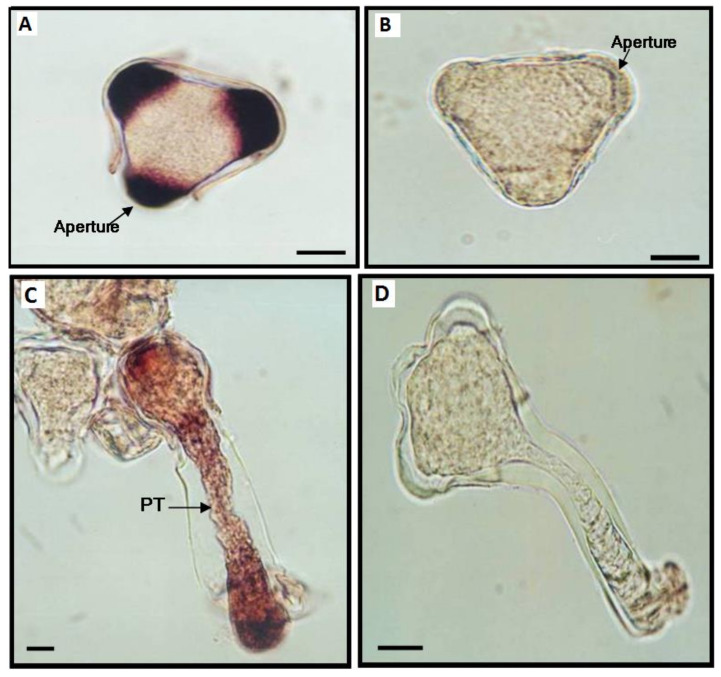
Prunus necrotic ringspot virus (PNRSV) RNA localization by whole mount in situ hybridization. (**A**) Viral RNA (purple color) is specifically localized in fully mature pollen grains at their apertures. (**B**) Uninfected pollen grain (negative control of the in situ hybridization) showing no purple color. (**C**) Infected, germinated pollen grain showing the PNRSV RNA (purple color) present not only inside the pollen grain but also inside the pollen tube (PT), especially on the tip. (**D**) Healthy, germinated pollen grain showing no positive signal after carrying out the whole mount in situ hybridization. Bar = 10 μm. This figure is adapted with permission from [14].

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
