# Peer review of "Imaging Techniques to Study Plant Virus Replication and Vertical Transmission"

_viruses, 2021, doi:10.3390/v13030358_

Round 1

Reviewer 1 Report

The manuscript has reviewed recent studies of application of different image techniques for studying plant virus replication and vertical transmission. The manuscript is well organized and well written. Here are some suggestions:

1, Line230, the statement needs further revision. This is maybe true for potexviruses, but for potyviruses, the virus replication complex is formed by 6K2, and other viral proteins.

2, Potyvirus as one of the biggest (+)ssRNA virus genus, there are quite a lot of studies on virus replication and movement. I suggest the author could add one section to review current research progress.

3, line 56, microcopy, misspelled? please revise.

Author Response

The manuscript has reviewed recent studies of application of different image techniques for studying plant virus replication and vertical transmission. The manuscript is well organized and well written. Here are some suggestions:

1, Line230, the statement needs further revision. This is maybe true for potexviruses, but for potyviruses, the virus replication complex is formed by 6K2, and other viral proteins.

We understand that the reviewer refers to the statement “vRNPs are the key players of the viral infectious cycle, driving various processes such as translation, transcription and intracellular trafficking. The replication process of ss (+) RNA viruses is carried out in virus-specific replication complexes (VRCs) formed by the RNA dependent RNA polymerase (RdRp; viral replicase), the replication-auxiliary proteins from the virus, vRNAs, and factors from the host, which…”

This statement is rather generic, and includes potyviral 6K2 or carmoviral p29 within the concept “replication-auxiliary proteins”. The triggering of the VRC formation is described in specific paragraphs, including a new section on potyvirus replication complexes in the revised version of the manuscript. Thus, we prefer to keep the statement as it was in the first version of the manuscript.

2, Potyvirus as one of the biggest (+)ssRNA virus genus, there are quite a lot of studies on virus replication and movement. I suggest the author could add one section to review current research progress.

The referee is right. We have followed the recommendation of the referee and added a new section on potyvirus replication factories.

3, line 56, microcopy, misspelled? please revise.

We found the misspelling and corrected it.

Reviewer 2 Report

The review article by Pina et al. gives a broad overview of using microscopy for visualizing plant virus-infected samples. The review clearly reviews the different options for imaging virus-infected samples and how they have been used to study plant virus replication. The article gives an in-depth review of studies examining VRCs from tombusviruses to potexviruses. The review is very well-written and easy to follow. This reviewer only offers the following minor suggestions to enhance this article. 

Comments:

It would be useful to include a figure depicting the tissue-printing hybridization for plant tissues infected with a plant virus. Visualizing the technique and results would greatly enhance the readers understanding of this method. Lines 39-51.

Figure 1: The arrowheads in panels C and E are difficult to find. It is also unclear what "505" refers to in the names EU_505 and CH2_505.

Lines 143-161. It would be useful to describe the most common combinations of fluorophores used in plant virology. e.g, Is GFP + RFP a popular combination? What combinations should be avoided? What tags are problematic on account of autofluorescence? This paragraph as-is does not describe in enough detail what problems are encountered using CSLM in plants.

Lines 257-259: It is stated that our current knowledge of the 3D architecture of virus replication factories is limited. Have viral replication factories been observed at high resolution in mammalian systems? If so, what technologies were most effective?

Figure 3 Legend: Is panel B showing an infected cell mitochondria? Is panel C uninfected? The legend is unclear.

Lines 384-385: What viral protein is fused to GFP? It is unclear in Fig. 4 what viral protein is being examined. Please describe the role(s) of the GFP-tagged protein in the virus lifecycle. 

Author Response

Reviewer 2:
The review article by Pina et al. gives a broad overview of using microscopy
for visualizing plant virus-infected samples. The review clearly reviews the
different options for imaging virus-infected samples and how they have
been used to study plant virus replication. The article gives an in-depth
review of studies examining VRCs from tombusviruses to potexviruses. The
review is very well-written and easy to follow. This reviewer only offers the
following minor suggestions to enhance this article.
Comments:
It would be useful to include a figure depicting the tissue-printing
hybridization for plant tissues infected with a plant virus. Visualizing the
technique and results would greatly enhance the readers understanding of
this method. Lines 39-51.
We have followed the recommendation of the referee and added a new
figure (current Figure 1) on tissue-printing hybridization.
Figure 1: The arrowheads in panels C and E are difficult to find. It is also
unclear what "505" refers to in the names EU_505 and CH2_505.
We have followed the recommendation for this figure, which is now Figure 2
and have changed the black into red arrowheads as well as referred to the
PepMV clones as EU and CH2.
Lines 143-161. It would be useful to describe the most common
combinations of fluorophores used in plant virology. e.g, Is GFP + RFP a 
popular combination? What combinations should be avoided? What tags
are problematic on account of autofluorescence? This paragraph as-is does
not describe in enough detail what problems are encountered using CSLM
in plants.
We have followed the recommendation and added a new paragraph
explaining that.
Lines 257-259: It is stated that our current knowledge of the 3D architecture
of virus replication factories is limited. Have viral replication factories been
observed at high resolution in mammalian systems? If so, what
technologies were most effective?
Along the manuscript we included some sentences and citations making
reference to mammalian and yeast systems which are ahead of plant
systems and can provide nice examples to follow. While this is clearly
illustrative of what can be done, there are many specificities in plants that
make them methodologically different and, at least at times, less accessible.
Therefore, we prefer to keep references to non-plant systems as they were
in the first version of the manuscript.
Figure 3 Legend: Is panel B showing an infected cell mitochondria? Is panel
C uninfected? The legend is unclear.
We have followed the recommendation and added a new sentence to clarify
it.
Lines 384-385: What viral protein is fused to GFP? It is unclear in Fig. 4
what viral protein is being examined. Please describe the role(s) of the
GFP-tagged protein in the virus lifecycle.
We have followed the recommendation and decide to add a new figure
(current Figure 3) to clarify this aspect. 

Reviewer 3 Report

I became increasingly excited the more I read of this manuscript. I think it is a very nice contribution. There are a few English language corrections that need to be made. 

For the Carmoviruses, is there some sort of signal that localizes virus genome, CP, etc to the mitochondria? How is the auxiliary replicase targeted; is there a localization signal? 

Gorgeous images, this paper when published will inspire young scientists to become plant virologists. 

Author Response

Reviewer 3:

Comments and Suggestions for Authors

I became increasingly excited the more I read of this manuscript. I think it is a very nice contribution. There are a few English language corrections that need to be made. 

For the Carmoviruses, is there some sort of signal that localizes virus genome, CP, etc to the mitochondria? How is the auxiliary replicase targeted; is there a localization signal? 

We have followed the recommendation and decide to add a new paraghraph on this subject.

Gorgeous images, this paper when published will inspire young scientists to become plant virologists.

Thanks a lot, this was indeed our intention, hopefully you are right!